# A New Root and Trunk Rot Disease of Grapevine Plantlets Caused by *Fusarium* in Four Species Complexes

**DOI:** 10.3390/jof11030230

**Published:** 2025-03-17

**Authors:** José Manoel Ferreira de Lima Cruz, Otília Ricardo de Farias, Brunno Cassiano Lemos Araújo, Alejandra Valencia Rivera, Cláudia Rita de Souza, Jorge Teodoro de Souza

**Affiliations:** 1Department of Plant Pathology, Federal University of Lavras, Lavras 37200-900, MG, Brazil; jose.cruz@estudante.ufla.br (J.M.F.d.L.C.); otiliarfarias@gmail.com (O.R.d.F.); brunno.lemos.ara@hotmail.com (B.C.L.A.); 2Faculty of Agricultural Sciences, Jaime Isaza Cadavid Colombian Polytechnic, Medellín 050022, ANT, Colombia; valencia.avr03@gmail.com; 3Technological Centre of Grape and Wine Research, Agricultural Research Agency of the State of Minas Gerais, Caldas 37780-000, MG, Brazil; crsouza@epamig.br

**Keywords:** aetiology, endophytic colonisation, genetic diversity, grapes, pathogenicity, *Vitis* spp.

## Abstract

Grapevines are propagated by grafting, but the rootstocks used in commercial plantations are susceptible to several diseases. In this study, we focused on a novel root and trunk rot disease of grapevine plantlets that show symptoms during cold storage, before field establishment. Our objectives were to study the aetiology, symptomatology, plant resistance responses, and mode of action of the pathogen that was initially identified as *Fusarium*. The characterisation of this pathosystem was performed by isolation, pathogenicity assays, genetic diversity studies with BOX-PCR, and identification by sequencing a fragment of the *tef1* gene. Scanning electron microscopy and X-ray spectroscopy were used to study the mode of action and plant resistance responses. The results showed that 12 species of *Fusarium*, initially isolated from both healthy and diseased plantlets, and classified into 4 species complexes, were pathogenic to grapevines. Comparative analyses between diseased and healthy roots showed typical resistance responses in diseased plantlets, including tyloses formation, translocation of Ca, and accumulation of Si. Field experiments confirmed that 100% of the diseased plantlets died within 90 days of transplantation. This study contributes to a better understanding of root and trunk rot disease under cold storage and provides insights for the development of management strategies.

## 1. Introduction

Grapevines (*Vitis vinifera*) are economically and socially important as they offer a wide range of products, including fresh and dried fruits, juices, liquors, and wines [1]. In Brazil, the production of grapes for fine wines is concentrated in the southern and southeastern regions, and the south of Minas Gerais stands out for both the expansion of cultivation and the high quality of the grapes and wines produced [2,3]. This performance is directly related to the favourable climate conditions allied with vineyard management practices, such as the production of high-quality plantlets (planting material) and the double pruning system, with a direct impact on the quality [2,3,4,5,6,7].

Production of grapevine plantlets is a long process that takes more than a year in Minas Gerais state, Brazil. The production of grafted grapevine plantlets is carried out using the table grafting system, in which the rootstock and scion of commercial cultivars are mechanically grafted. These unrooted plantlets undergo a stratification for the formation of callus at the junction between the scion and rootstock for a period of approximately 20 days at 20–29 °C and 90–95% humidity. After this period, the unrooted plantlets are transferred to a nursery installed in the field, where they remain for one year for root development. After root development, the rooted plantlets are removed from the soil, cleaned, have their roots trimmed, and are stored in a cold chamber at 4 °C for periods that may go up to one year before commercialisation [8].

The Agricultural Research Agency of Minas Gerais State (EPAMIG) is one of the institutions responsible for producing grapevine plantlets to expand grape cultivation in Minas Gerais. Plantlets with rotten roots and trunks were first observed in 2016 at the plantlet producing facility coordinated by EPAMIG in the Caldas municipality, Minas Gerais. Preliminary observations suggested that *Fusarium* was associated with these symptoms. This root and trunk rot disease was more prevalent in plantlets with the hybrid rootstocks IAC 313, IAC 572, IAC 766, and 1103P, widely used due to their high vigour and resistance to phylloxera aphids [6,9,10]. Initial observations indicated that root and trunk rot symptoms only developed under cold storage. An informal quantification of the incidence of root and trunk rot indicated that up to 50% of the susceptible plantlets could develop symptoms (Souza, C.R., personal observation). Since its discovery, little research has been undertaken to understand the aetiology and control of this disease.

It is well known that grapevines are susceptible to several diseases, including grapevine trunk decline (GTD), certainly the most complex and one of the most destructive diseases of this crop. GTD shows symptoms in the field, typically after 5 years of cultivation, but varies according to a great number of factors [11,12]. The disease may cause between 30 and 94% of losses and involves a great number of fungal pathogens in its aetiology [13]. The symptoms involve root rot, trunk rot, and general decline, which may vary according to the causative agent. More than 140 fungal species were listed as causative agents, including *Neofusicoccum*, *Phaeomoniella*, *Diplodia*, *Lasiodiplodia*, *Botryosphaeria*, *Phaeoacremonium*, *Pleurostoma*, *Cadophora*, *Campylocarpon*, *Cylindrocladiella*, *Dactylonectria*, *Ilyonectria*, *Neonectria*, and *Thelonectria* [11,12,14,15,16,17,18,19,20]. *Fusarium* is among the agents of GTD, but it is always in association with other pathogens [21]. *Fusarium* is a pathogen with a wide host range, frequently acting as an opportunist, infecting plants under stressful conditions [22,23]. *Fusarium* was recently reported as the causative agent of trunk rot in plantlets in Turkey [22].

In this study, we report a new disease caused by *Fusarium* that we named root and trunk rot of grafted grapevine plantlets under cold storage, and it is clearly different from GTD. In this initial characterisation of the disease, we described the symptoms, identified the *Fusarium* species associated with healthy and diseased plantlets, and studied the reactions of the plant to the pathogens. The factors that are thought to influence disease development and possible control strategies are discussed.

## 2. Materials and Methods

### 2.1. Plant Material and Fungal Isolation

Healthy and diseased grapevine plantlets of cv. Syrah (*Vitis vinifera*) grafted onto IAC766 rootstock ((*Vitis riparia* × *Vitis cordifolia*) × *Vitis caribea*) were obtained from the plantlet production areas of the EPAMIG research station, in Caldas, Minas Gerais, Brazil. The samples were placed in plastic bags and stored at 4 °C until fungal isolation. A total of 25 healthy and 25 diseased plantlets of grafted cv. Syrah on cv. IAC766 were used in these isolations. In the laboratory, tissue fragments (diam = 5 mm) from the trunk and roots of healthy and diseased plantlets were surface sterilised by immersion in 70% ethanol for 30 s and 1% NaClO for 60 s, washed three times with sterile distilled water, and transferred to Petri dishes containing potato dextrose agar (PDA) medium with streptomycin (250 mg/L) for 7 days at 25 °C, with a 12 h photoperiod. Subsequently, the fungal structures that developed on the fragments were purified and preserved in sterile distilled water [24] and glycerol (40%) at −80 °C. Fungal strains were identified at the genus level based on the morphology of the reproductive structures observed under a microscope and comparison with descriptions in the specialised literature [25,26].

### 2.2. Clustering and Identification

Strains of *Fusarium*, selected for being the most prevalent genus in both healthy and diseased plantlets, were subjected to total genomic DNA extraction following the procedures described by [27]. The purity of the extracted DNA was estimated, and its concentration was quantified using a spectrophotometer (Nanodrop Lite UV-Vis; Thermo Fisher Scientific, San Francisco, CA, USA). The DNA was then stored at −20 °C for further analyses.

BOX-PCR, a reproducible whole-genome fingerprinting technique used here to study the genetic diversity of the *Fusarium* strains, was performed in 25 μL reactions with the primer BOX-A1R (5′-CTA CGG CAA GGC GAC GCT GAC G-3′) [28]. Cluster analysis was performed using the unweighted pair group method using arithmetic averages (UPGMA) hierarchical method in R statistical software version 4.4.2 [29]. Genetic similarity was assessed based on Jaccard’s distance [30] and calculated from a binary matrix indicating the presence (1) or absence (0) of amplified products (bps).

*Fusarium* strains obtained from healthy plantlets, as well as representative strains of each BOX group identified through cluster analysis, were selected for sequencing a fragment of the *tef1* gene (translation elongation factor 1-α). Polymerase chain reaction (PCR) amplification of the gene was performed using the primers EF-1 (5′-ATG GGT AAG GAR GAC AAG AC-3′) and EF-2 (5′-GGA RGT ACC AGT SAT CAT GTT-3′) following the recommendations of [31]. Amplified fragments were purified using the commercial Biospin PCR kit and sequenced in a private facility (ACTgene Molecular Analysis, Porto Alegre, RS, Brazil). Consensus sequences were obtained by assembling and editing in Sequencher v. 5.4.6 (Gene Codes Corporation, Ann Harbor, MI, USA) and compared with the most closely related sequences of type material deposited in public databases [32] using the BLASTn programme (https://blast.ncbi.nlm.nih.gov/Blast.cgi, accessed on 20 November 2024) [33]. Phylogenetic analyses using the *tef1* sequences obtained and sequences from the closest type material deposited in databases (Appendix A) were performed to identify the strains at the species level. Alignments were conducted using MAFFT (multiple alignment using fast Fourier transform) v. 7 [34,35] and refined using GBlocks v. 0.91b [36], eliminating low-quality regions and preserving only the conserved blocks. Evolutionary relationships were inferred in RAxML-HCP2 v. 7.0.4 and MrBayes on Access (3.2.7a) [37], implemented in the CIPRES platform [38], using maximum likelihood (ML) and Bayesian inference (BI) methods separately, with 1000 bootstrap replications. Final phylogenetic trees were edited using FigTree v. 1.4.3 [39] and *tef1* sequences were deposited with GenBank.

### 2.3. Pathogenicity and Virulence

Three inoculation methods were performed in parallel using *Fusarium* strains on cv. IAC766 grafted with cv. Syrah, (i) trunk inoculation, (ii) inoculation of rooted plantlets, and (iii) inoculation of the base of rootless plantlets [22,40].

For method (i), mycelium discs (3 mm diameter) from each strain, previously grown on PDA plates, were used. The area between the base and the first node of the plantlets was punctured with a sterile cork borer to insert the inoculum. PDA discs without mycelium were used as negative controls. To prevent water loss, sterile cotton soaked in distilled water was placed on the opposite side of the lesion. Pathogenicity is defined here as the capacity of a pathogen to cause disease and virulence as the amount (incidence in this case) of disease, which was quantified with a descriptive scale (Table 1). The pathogenicity of representative strains from each BOX-PCR cluster was tested following method (i). Plants were examined under a stereomicroscope to visualise disease symptoms and pathogen signs at 7 days after the inoculations. The virulence was determined on the basis of the number of plantlets with trunk rot incidence using the scale described in Table 1. Nine strains were chosen to complete Koch’s postulates by performing re-isolations on PDA. The assays with method (i) were conducted in a completely randomised design with 15 plantlets per treatment.

Methods (ii) and (iii) consisted of inoculations with conidial suspensions prepared by growing the fungi in liquid malt extract medium for 7–10 days at 25 °C, subsequently filtering the culture through a double layer of gauze and adjusting the concentration to 10^7^ conidia/mL in a Neubauer chamber. For method (ii), rooted plantlets that were produced in field nurseries were immersed in the suspensions of each *Fusarium* strain for 24 h and stored in a cold chamber for 3 months. For method (iii), rootless plantlets (without roots) were inoculated as described above, planted in the nursery field for 12 months, and subsequently, the plantlets were removed from the soil, cleaned, and stored in a cold chamber (4 °C) for 3 months, following standardised methods described in [8]. The experiments with methods (ii) and (iii) were installed in a randomised block design with four replicates of 25 plantlets per replicate, totalling 100 plants per treatment. The fungicide treatment consisted of immersing the plantlets for 24 h in a solution containing 200 mL/100 L of water of a commercial product composed of 50% of the active ingredient Iprodione. The survival rate of the plantlets and disease incidence were evaluated after 3 months of storage in the cold chamber. Later on, 100 healthy and 100 diseased plantlets, without discriminating the *Fusarium* strains in diseased plantlets, were transplanted to the field to assess their survival rate. Mortality, shoot length, and chlorophyll a, b, and total chlorophyll content were measured using ClorofiLOG model CFL 1030 (Falker, Porto Alegre, RS, Brazil). These measurements were performed on 100 healthy and 100 diseased plantlets growing in the field by recording chlorophyll measurements in three leaves per plant. The trials were conducted with the five most virulent strains among those that fulfilled Koch’s postulates performed with method (i). All analyses were performed using the R software v.4.3.3 [29], with the data subjected to residual normality tests using Shapiro–Wilk and the variance homogeneity test of Bartlett. Differences between the means of healthy and diseased plantlets were analysed using the *t*-test 5 and 1% probability.

### 2.4. Pathogenicity and Colonisation of Other Plant Species by Fusarium

In order to evaluate whether *Fusarium* strains from diseased plantlets were capable of colonising and causing root and/or trunk rot in other crops, conidial suspensions containing 10^7^ conidia/mL were used to inoculate common bean cv. Carioca and tomato cv. Santa Cruz seedlings, and grapevine rooted plantlets of cv. Syrah grafted onto cv. IAC766. Grapevine plantlets were rooted in a mixture (1:1) of sterile sand and a substrate (Tropstrato, Campinas, SP, Brazil) in a greenhouse for approximately 1 month before the inoculations. The roots of 25 plants of each species were surface sterilised as described above, immersed in conidial suspensions of each strain that had fulfilled Koch’s postulates, and incubated for 7 days in Petri dishes covered with filter paper moistened with sterile distilled water to maintain the humidity. After the incubation period, disease incidence was visually assessed and roots were cut in 0.5 cm-long fragments, surface sterilised as described in Section 2.1, and then 100 fragments per plant species per *Fusarium* strain were plated on PDA to quantify root colonisation. The fungal colonies obtained were visually compared with the originally inoculated strains.

### 2.5. Comparative Analysis of Diseased vs Healthy Tissues

Scanning electron microscopy (SEM) and X-ray spectroscopy were used to assess the interaction between pathogenic *Fusarium* strains and grapevine plantlets. The study included a structural analysis and the determination of the chemical composition of both healthy and diseased plantlets.

Fragments from roots and trunks of healthy and diseased plantlets were fixed in a modified Karnovsky solution (2.5% glutaraldehyde, 2.5% paraformaldehyde, 0.05 M sodium cacodylate buffer, pH 7.2, and 0.001 M CaCl_2_), following the protocol described in [41]. The samples remained in this solution for 24 h, then were washed with cacodylate buffer (0.05 M) and subjected to sequential dehydration in acetone at concentrations of 25%, 50%, 70%, and 90% for 10 min each, followed by 100% acetone three times for 10 min each. After dehydration, the samples were dried at the critical point using CO_2_ (Baltec, model CPD 030; Electron Microscopy Sciences, Hatfield, PA, USA). The specimens were mounted on aluminium stubs (covered with aluminium foil) using double-sided carbon tape and coated with gold (Balzers SCD 050, Oerlikon Balzers, Balzers, Liechtenstein). The prepared samples were stored in a desiccator with silica gel until SEM analysis (model Leo Evo 40 XVP, Carl Zeiss, Jena, Germany).

For chemical composition analysis, the fragments were directly fixed on aluminium stubs for 24 h in a desiccator containing silica gel for dehydration. The samples were then coated with carbon in a sputtering machine (MED 010, Oerlikon Balzers, Balzers, Liechtenstein). Mineral element mapping was performed using energy-dispersive X-ray spectroscopy (EDS), integrated with the SEM, using Bruker Espirit software (version 1.9). Images were generated and digitally captured under the working conditions of 20 kV, at 9.0 mm working distance, ±180× magnification, and Kcps values ranging between 3 and 4. The nutrients that showed variations in distribution between diseased and healthy plantlets were selected for graphical representation. The images from the morphological analysis and the chemical mapping results were processed and edited using CorelDraw and CorelPhoto-Paint v. 22.0.

## 3. Results

### 3.1. Symptoms of Root and Trunk Rot of Grapevine Plantlets Under Cold Storage

The symptoms of root and trunk rot in grapevine plantlets only appear during storage in a cold chamber at 4 °C. Root and trunk rot under cold storage was first detected in cultivar Syrah grafted onto IAC766 rootstock and since 2016 its occurrence has been noticed in the plantlet-producing facility of EPAMIG in Caldas, Minas Gerais state. However, we were not able to quantify its incidence precisely as it varies greatly from year to year, but it currently affects roughly around 5–10% of the plantlets produced annually (C.R. Souza, personal observation). Diseased plantlets showed reduced vigour when they are grown in a greenhouse, and yellowing and leaf fall is quickly followed by plant death (Figure 1A,B). The disease progresses from the rotten roots (Figure 1C), reaches the crown and trunk, and finally the callus formed at the union between the scion and the rootstock (Figure 1D–F). The internal tissues of the trunk show darkening of the xylem vessels accompanied by areas of rot that appear as dark longitudinal bands (Figure 1E,F).

Diseased plantlets sprouted when transferred to the field and produced shorter shoots, but 100% of them (n = 100) died in a period of 90 days after field transplantation, whereas no death of healthy plantlets was observed in this period. Healthy plantlets produced significantly longer and more vigorous shoots that were on average 21 cm long, in contrast to diseased plantlets that produced shoots 8.2 cm long on average (Figure 2A–C). Healthy plantlets showed significantly higher levels of chlorophyll than diseased plantlets. These values were approximately 1.8, 2.8, and 2× higher in healthy than in diseased plantlets for chlorophyll A, chlorophyll B, and total chlorophyll, respectively (Figure 2D). Furthermore, healthy plantlets exhibited more developed branches and leaves with an intense and uniform green colour (Figure 2B,C,E,F).

### 3.2. Fusarium Was the Most Prevalent Genus in Both Healthy and Diseased Plantlets

In order to identify the pathogens associated with root and trunk rot of grapevine, healthy and diseased plantlets of cv. Syrah grafted onto rootstock IAC766 were subjected to the isolation of microorganisms in culture media. The genera *Fusarium* and *Epicoccum* were the most commonly isolated genera from the roots of both diseased and healthy plantlets, as identified by morphological analysis (Figure 3). Bacteria were also isolated, but their numbers were considered low and, therefore, our attention turned to the fungi.

A total of 156 strains were analysed from healthy plantlets, from which *Fusarium* represented 19%, whereas 236 strains were analysed from diseased plantlets, with *Fusarium* representing 39% of the total.

### 3.3. Fusarium Strains from Diseased Plantlets Were Highly Diverse

The genus *Fusarium* was the most prevalent in the roots of both healthy and diseased plantlets and therefore was the focus of our diversity studies. A total of 86 strains of *Fusarium* from diseased plantlets were randomly selected for a diversity study with BOX-PCR fingerprinting. These strains were grouped into 45 genetic groups on the basis of 53 polymorphic bands. From these genetic groups, 10 were composed of more than one strain, whereas 35 were formed of one unique strain (Figure 4).

In order to identify the strains of *Fusarium* from diseased plantlets, at least one strain from each BOX group was selected for the sequencing of a fragment of the *tef1* gene. A total of 78 strains were used in these identifications, 49 from diseased plantlets and 30 strains isolated from healthy plantlets. The strains from diseased plantlets included 45 from each BOX group and 5 other strains from BOX groups containing more than 1 strain (Figure 4). Sequence and phylogenetic analyses allowed the identification of 14 *Fusarium* species among the 49 strains from the roots of diseased plantlets (Figure 5). These 14 *Fusarium* species belong in four species complexes as follows: 23 strains in the *Fusarium oxysporum* species complex (FOSC), 21 strains in the *F. fujikuroi* species complex (FFSC), 3 strains in the *F. tricinctum* species complex (FTSC) and 2 strains in the *F. solani* species complex (FSSC) (Table 2, Figure 5). These species were identified by comparing their *tef1* sequences with the type strains deposited in public databases (Appendix A) and most of them were 98.47–100% identical. However, some strains, such as LMF0441, LMF0442 and LMF0427 that were closest to *F. chongqingense*, and LMF0417, LMF0420, LMF0414, LMF0421, LMF041, LMF046 and LMF0432 that were close to *F. foetens*, showed identities of 97.48–98.2% when compared to type strains deposited in databases. Although we have named them based on their closest related sequences they may represent novel *Fusarium* species.

The 30 strains isolated from healthy grapevine plantlets were not analysed by BOX-PCR and all of them were identified by sequencing a fragment of the *tef1* gene. These 30 strains were identified as five species in three species complexes, 15 strains in the FFSC, 14 strains in the FOSC and 1 strain in the FIESC (Figure 5).

The species *F. casha* and *F. fredkrugeri* in the FFSC and *F. vaughaniae* and *F. inflexum* in the FOSC occurred in both healthy and diseased plantlets, whereas *F. brevicaudatum* was only obtained from healthy plantlets (Figure 5). Some strains identified in the same species were classified in more than one BOX group, but there were no cases where strains in the same BOX group were identified in different species (Figure 4 and Figure 5).

### 3.4. Fusarium from Diseased Plantlets Are Pathogenic to Grapevine

In this study, three methods were used to evaluate the pathogenicity and virulence of the *Fusarium* strains to grapevines. The trunk inoculation method was adopted for at least one strain per BOX group, whereas the inoculation at the base of rootless plantlets and on rooted plantlets were used for a limited number of strains (see below). From a total of 49 strains obtained from diseased plantlets and classified into 13 species in 4 species complexes, 12 species were able to cause trunk rot in grapevine plantlets (Table 2). These 12 pathogenic species are distributed in 43 BOX groups, from a total of 45 BOX groups obtained in the study (Table 2). There were no relationships between virulence and BOX group or phylogenetic identity of the strains (Table 2). Non-inoculated plantlets showed no trunk rot symptoms.

To reproduce the symptoms of root and trunk rot, the trunk, and rooted and unrooted plantlets were inoculated with randomly selected strains (Figure 6). The inoculation of rooted plantlets followed by storage at 4 °C for four months did not result in any disease symptoms. On the other hand, the inoculation of the trunk resulted in trunk rot by all strains at various degrees (Figure 6A). These nine strains were also re-isolated from the diseased plantlets and morphologically compared to the inoculated strains, completing Koch’s postulates. The inoculation of the base of unrooted plantlets followed by 1 year of growth in the field and 3 months of storage at 4 °C resulted in root and trunk rot by the five most virulent strains (Figure 6B) as determined in the trunk inoculation experiment (Figure 6A). Rooting in the field nursery was not influenced by the inoculation of the different *Fusarium* strains. The disease was not observed after the removal of inoculated plantlets from the soil where they were cultivated for one year for rooting; therefore, cold storage was essential for disease development.

There was a correspondence between the incidence of vascular trunk rot (Figure 6A) and root rot after cold storage (Figure 6B). The fungicide treatment of the unrooted plantlets was effective in inhibiting the pathogen and the non-inoculated negative control did not show disease symptoms (Figure 6).

### 3.5. Comparative Analysis of Healthy and Diseased Tissues

Further studies with scanning electron microscopy (SEM) and X-ray spectroscopy were used to understand the relationship between plant and pathogen. The SEM images showed that healthy plantlets had normal xylem vessels in roots (Figure 7A,B), whereas diseased plantlets had xylem vessels clogged with tyloses and gum (Figure 7C,D). *Fusarium* hyphae was observed in diseased tissues (Figure 7E,F), but not in healthy vascular tissues (Appendix A).

Energy-dispersive X-ray spectroscopy (EDS) analyses showed differences in the mineral composition of healthy and diseased plantlet roots. Healthy root tissues had lower levels of calcium (Ca), aluminium (Al), and magnesium (Mg), and had higher levels of potassium (K) when compared to diseased root tissues (Figure 8A,B). In healthy roots, Ca was more evenly distributed and part of it was concentrated in the epidermis (Figure 8C), whereas in diseased plantlets it was in the xylem vessels (Figure 8D). Silicon was not detected in healthy tissues but was uniformly distributed in diseased tissues (Figure 8E,F).

### 3.6. Fusarium Colonises Roots of Different Species, but Does Not Show Symptoms

*Fusarium* strains obtained from diseased plantlets and previously shown to cause root and/or trunk rot (Table 2, Figure 6) were inoculated in grapevine roots and in tomato and common bean seedlings by root immersion to verify if they could colonise these plants and cause symptoms (Table 3). The results showed that all strains were re-isolated from the surface-sterilised roots of all plant species without causing any symptoms (Table 4). The strains were from six BOX groups, six species, and three species complexes, indicating that this is a common behaviour in a diverse range of *Fusarium* species. The percentages of re-isolations were high for all strains in all plant species and the non-inoculated plantlets showed no contamination (Table 3).

### 3.7. Fusarium Strains from Healthy Plantlets Are Also Pathogenic to Grapevine

Strains of *Fusarium* obtained from the roots and trunks of healthy plantlets were inoculated in the trunk of grapevines to study their capacity to cause disease. None of the strains were isolated from the leaves and petioles of healthy plantlets. Interestingly, all strains in the FOSC were from the roots, while strains in the FFSC were from the roots and from the trunks (Table 4). The strains from healthy plantlets were identified as five species in three species complexes (Table 4). The identity of the *tef1* fragment obtained for the strains from healthy plantlets varied from 98.6% to 100% when compared to the sequences of type strains deposited in public databases. The results showed that most strains were able to cause trunk rot in grapevines. *Fusarium* strains from the FFSC and FOSC were able to cause disease, whereas *F. brevicaudatum* in the FIESC was not pathogenic (Table 4). Strains from the same species varied in their pathogenicity and also in the amount of disease symptoms in grapevines (Table 4).

## 4. Discussion

In this study, we report a novel root and trunk rot disease of grapevine plantlets under cold storage caused by *Fusarium* species. This disease only shows symptoms when the plantlets are exposed to the cold after being cultivated with the pathogen for a whole year in the commercial process of producing grafted grapevine plantlets. This disease invariably leads to plant death within a period of 90 days after field transplantation. There were 14 different *Fusarium* species in four species complexes associated with healthy and diseased plantlets. Of these 14 species, 12 were able to cause disease and 9 are being reported for the first time as pathogenic to grapevines. *Fusarium* species obtained from both diseased and healthy plantlets were pathogenic to grapevines, indicating that healthy infected plantlets can carry the pathogens that may later be responsible for plant decline in the field.

The *Fusarium* genus is commonly associated with grapevine trunk disease (GTD) in several countries as it is part of the fungal complex that causes it, but this genus is not considered to be the main causative agent of GTD [21,22,23,42,43,44,45,46]. In our study, *Fusarium* was the only fungal genus considered in the pathogenicity assays due to the high populations found in healthy and diseased grapevine plantlets before transplantation to the field. Symptoms of root and trunk rot include a general loss of vigour and the vascular rot of the root and trunk of diseased plantlets. Despite the damage to the root system, sprouting occurred due to the accumulation of starch and sugar reserves in the plantlets [47,48]. Marked differences between these diseases are the rapid plant death (90 days) in root and trunk rot and the main role of *Fusarium*, whereas GTD has the involvement of a high number of pathogens, including *Fusarium* as a secondary pathogen, and it occurs in young orchards (<5 years) and develops slowly, allowing plants to grow and be productive for years before death [11,12,18,19,21,49]. Root and trunk rot under cold storage seems to be an overload of *Fusarium* on diseased plantlets that rapidly leads to plant death. Other pathogens than *Fusarium* were not detected in our isolations and further studies with deep sequencing (NGS) need to be performed to verify their association in plants with root and trunk rot symptoms. Moreover, it will be interesting to study whether plantlets infected with *Fusarium* without root and trunk rot symptoms may develop GTD after transplantation to the field.

Our pathogenicity tests were performed with three different methods, and only in the inoculation of unrooted plantlets followed by cultivation in the field nursery for one year and storage for 3–4 months at 4 °C the symptoms of root rot were observed. However, due to the length and cost of this process, we only tested five strains in three species classified in three species complexes. The inoculation of rootless plantlets method correlated well with the trunk inoculation method, which was used to test the pathogenicity of all *Fusarium* strains in this study. The trunk inoculation method was fast, reproducible, and easy to perform and became our standard pathogenicity test.

Nine *Fusarium* species, from a total of fourteen found in this study, were identified for the first time as pathogenic to grapevines, including *F. chongqingense* in the FTSC, *F. foetens*, *F. inflexum*, *F. landiae*, *F. vaughaniae*, *F. triseptatum*, and *F. inflexum* in the FOSC, and *F. fujikuroi*, *F. guttiforme*, and *F. casha* in the FFSC. The species *F. brevicaudatum* and *F. chinhoyiense* in the FIESC and FFSC, although not pathogenic in our assays, were also found for the first time in association with grapevines. These results demonstrated that the species associated with root rot and trunk disease in Minas Gerais state are diverse and distinct from the ones reported by other authors elsewhere [21,22,23,42,43,44,45,46]. Among the species reported in our study, the ones close to *F. foetens* and to *F. chongqingense* (Figure 5) may be novel species due to the relatively low levels of identity of their *tef1* gene as compared to type strains deposited in public databases. However, sequencing of other markers such as *rpb2* and calmodulin is required to corroborate this postulation.

The pathogenicity and virulence of the strains tested in our study varied considerably, even in strains from the same species. Some of these differences were detected in our BOX-PCR analysis (Figure 4), a fingerprinting technique that locates variations in the whole genome [28]. BOX-PCR is sensitive enough to detect differences among strains that are not detected when a single fragment of DNA, such as *tef1*, is analysed. Therefore, there was no relationship between our phylogenetic placement (Figure 5) and BOX-PCR groupings (Figure 4). For example, strains LFM0410, LMF0412, and LMF0426 of *F. vaughaniae*, classified in three different BOX groups, markedly differed in pathogenicity and virulence to grapevines (Table 2). These differences are probably due to virulence factors that may be located in accessory chromosomes [50,51,52,53,54,55,56,57,58,59]. Alternatively, the loss and gain of pathogenicity and virulence in *Fusarium* may be due to horizontal transfer of genetic material through conidial anastomosis tubes [60,61] and subculture in laboratory artificial culture media [62].

The occurrence of *Fusarium* in healthy grapevine plantlets was reported by other authors [22,23,63,64], but their pathogenicity has not been confirmed. In our study, four *Fusarium* species, *F. casha*, *F. fredkrugeri*, *F. inflexum*, and *F. vaughaniae*, were isolated from healthy and diseased plantlets and their pathogenicity to grapevines has been confirmed with the trunk inoculation method (Table 2 and Table 4). These results show that *Fusarium*, like many other pathogens with a necrotrophic phase, may remain inside healthy tissues for long periods of time without showing any visible symptoms [21,65]. This endophytic colonisation occurs because the pathogen is unable to overcome the resistance of the plant and may remain incognito until the level of local resistance lowers [66]. Stresses such as the storage of plantlets with exposed roots to cold temperatures for 3–4 months are thought to be responsible for lowering the levels of resistance and the consequent manifestation of grapevine root and trunk rot symptoms. Experiments to confirm this hypothesis would involve the quantification of the resistance and colonisation levels, either by qPCR or enzymatic determinations [67].

Root and trunk rot of grapevine plantlets has similarities with the canker disease of cold-stored fruit and tree nuts [68], where plantlets were exposed to 5 to 10 °C for approximately 2 months. However, the *Fusarium* species inciting the disease was among the differences observed in our study. Despite the fact that old nomenclature still adopts *formae speciales* for some *Fusarium* species [69], there is increasing evidence that most species in this genus present very little host specificity and high levels of opportunism [70]. An example from our own results is *F. guttiforme*, which was found to be pathogenic to grapevine, but is only known for its association with pineapple fruit rot [71]. The high amount of *Fusarium* species associated with grapevine diseases was reported by other authors [21,22,23,42,43,44,45,46], and because of the broad host range of *Fusarium* species in general it appears that at least in some instances the local diversity of species influences the aetiology of any given disease incited by this genus. For example, grapevine trunk disease was reported to be caused by different *Fusarium* species according to the geographical region it is reported [42,44,46]. Our attempts to inoculate grapevines and other plant species with *Fusarium* spp. from diseased plantlets did not lead to root rot symptoms, instead, they resulted in extensive endophytic colonisation (Table 3). Further studies will be pursued by our research group to verify if different stresses would induce root rot symptoms in these plant species.

The rootstock IAC766 used in our pathogenicity tests is reported to possess a certain level of resistance against *Fusarium* spp. [72]. In fact, we observed the production of tyloses and gum in the xylem of diseased plantlets, translocation of Ca, and accumulation of Si (Figure 7 and Figure 8). These are typical manifestations of resistance [73,74,75,76,77,78,79,80,81] that were overcome by *Fusarium* under cold storage. An important requirement for disease development is the level of colonisation of the internal tissues by *Fusarium*, as suggested by [64,82]. In our study, *Fusarium* probably reached high levels of colonisation only when the inoculations were performed on the base of rootless plantlets. This occurred due to the approximately one year of plant growth in the field with the pathogen. By contrast, in inoculations performed on rooted plantlets, the pathogen had not enough time to grow together with the roots and reach high populations. This would explain why the disease only developed in inoculations performed in unrooted plantlets. Other factors that probably influence disease development are the composition and diversity of the internal and external microbiomes of the plantlets, their nutritional status and consequently their general resistance level, and the time in cold storage. Diverse and resilient microbiomes in well-nourished plantlets will restrict colonisation and favour resistance reactions against pathogens in general [83,84]. Further studies are needed to understand the influence of *Fusarium* colonisation levels, plant resistance, and the associated microbiomes in disease development. The presence of beneficial fungi, such as *Trichoderma*, *Epicoccum*, and *Chaetomium*, in addition to pathogens in our isolations (Figure 3) indicates that changing the population balance toward increased numbers of beneficial microorganisms is a promising strategy to manage this disease. Observations performed by one of us (de Souza, C.R., personal observations) indicated that shorter periods under cold storage led to a decreased incidence of root and trunk disease in susceptible material, which is currently around 5 to 10% in the last 5 years. The treatment with the fungicide, although effective in disease control, comes with all the disadvantages of these molecules, including environmental, human, and animal health hazards, and the risk of selecting resistant pathogen populations [85].

In conclusion, *Fusarium* species are present in both healthy and diseased plantlets, but the symptoms of the root and trunk rot of grapevines only appear under certain conditions, including cold storage and possibly a minimum threshold level of root colonisation. This study adds information on the grapevine root rot and trunk disease, including symptoms, diversity of the pathogen, identity and mode of action of the causative agents, disease development, and plant resistance reactions. This information will be important to develop control strategies against this disease.

## Figures and Tables

**Figure 1 jof-11-00230-f001:**
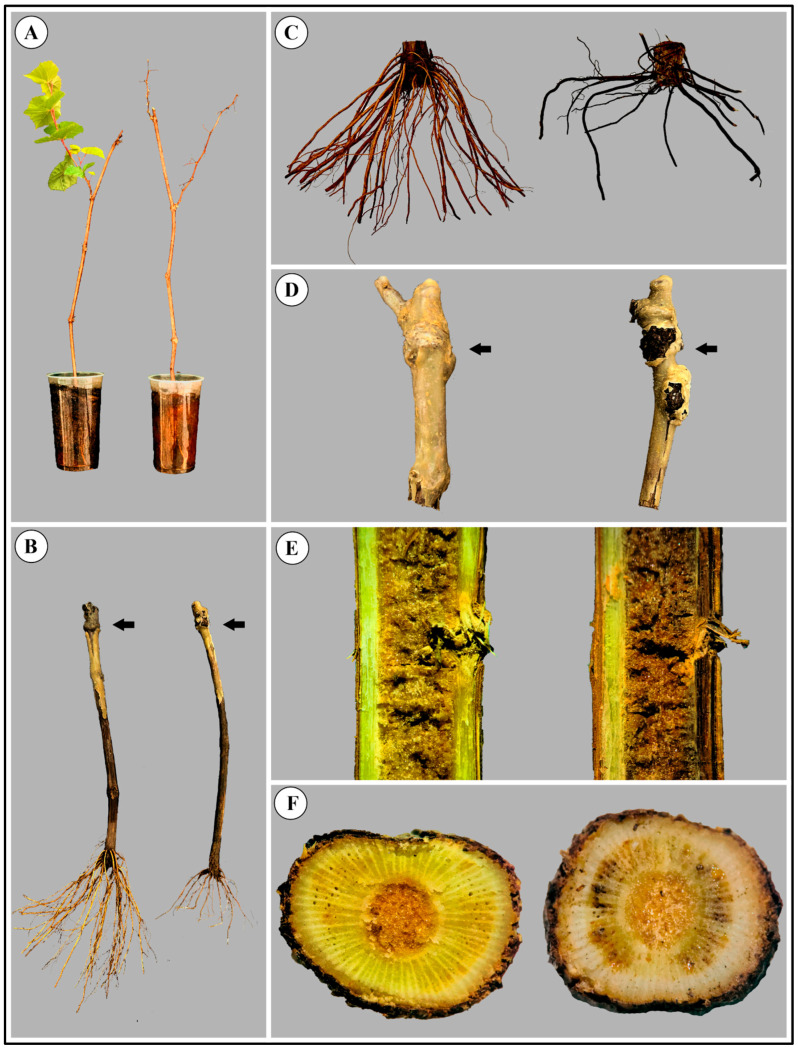
Symptoms of root and trunk rot in grapevine plantlets cv. Syrah grafted onto IAC766 under cold storage. Healthy plant material is always shown on the right side and diseased plant material on the left side. (**A**) Whole plantlet showing branching and diseased plantlet without leaves. (**B**) Normal roots on a healthy plantlet and rotten roots on a diseased plantlet. (**C**) Healthy and diseased roots. (**D**) Callus formed in the junction between the scion and the rootstock in healthy and diseased plantlets. (**E**) Longitudinal section in the vascular tissue of healthy and diseased plantlets showing darkening. (**F**) Cross sections in healthy and diseased vascular bundles of the trunk showing darkening due to the disease. Black arrows indicate the junction between scion and rootstock.

**Figure 2 jof-11-00230-f002:**
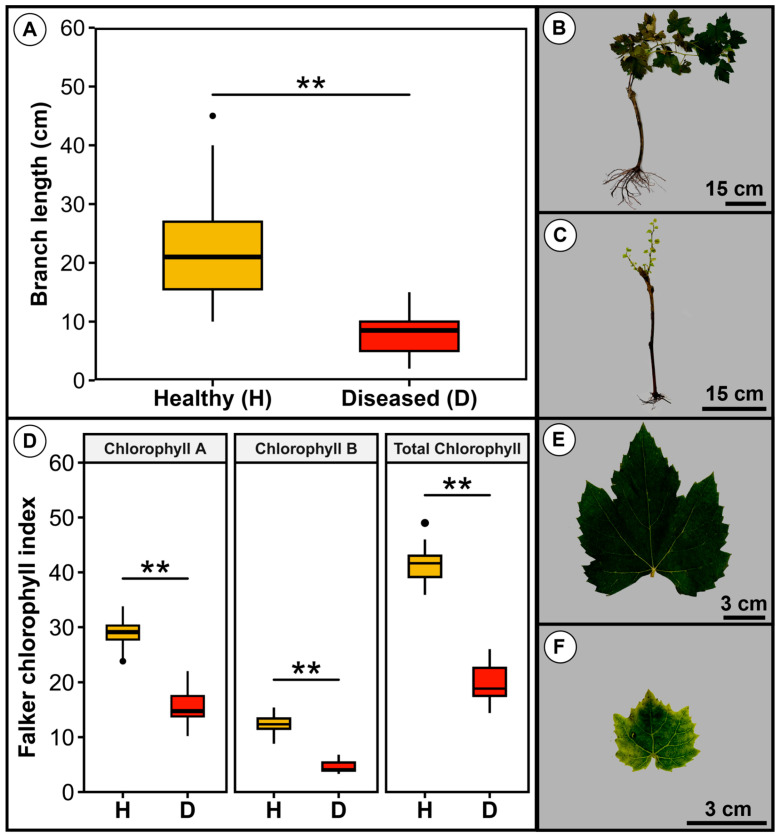
Sprouting and chlorophyll content in healthy—H and diseased—D plantlets of cv. Syrah grafted onto IAC766 rootstocks. (**A**) Branch length in 100 healthy and 100 diseased plantlets. (**B**) Visual aspect of a sprout in a healthy plantlet. (**C**) Sprouts in a diseased plantlet. (**D**) Chlorophyll content in 100 healthy and 100 diseased plantlets, with the measurement of 3 leaves per plantlet. (**E**) Aspect of a leaf in a healthy plantlet. (**F**) Leaf on a diseased plantlet. Asterisks indicate significant differences according to the *t*-test at 1% probability. Scale bars are shown at the lower right corner of figures in the right panel.

**Figure 3 jof-11-00230-f003:**
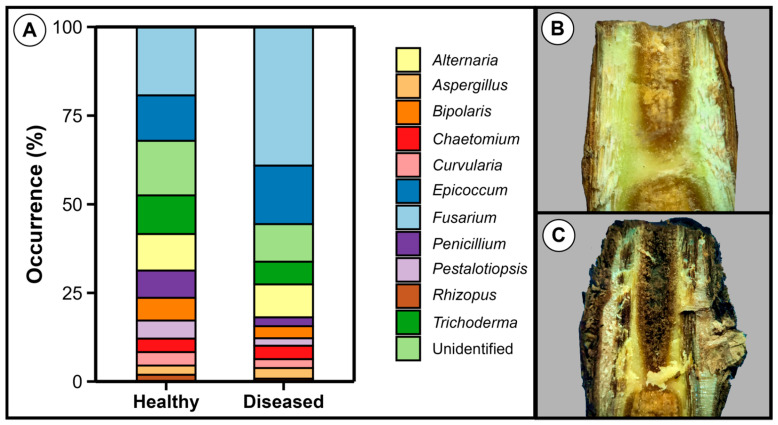
Microorganisms associated with healthy and diseased grapevine plantlets. (**A**) Occurrence of fungi in roots of grapevine cv. Syrah grafted onto rootstocks of cv. IAC766. (**B**) Visual aspect of the plantlet crown, defined as the area between the root and the trunk. (**C**) Crown of a diseased plantlet. A total of 25 healthy and 25 diseased plantlets of grafted cv. Syrah on cv. IAC766 were used in the isolations. Identifications were performed through morphological analysis using a microscope and the number of colonies was determined on PDA plates.

**Figure 4 jof-11-00230-f004:**
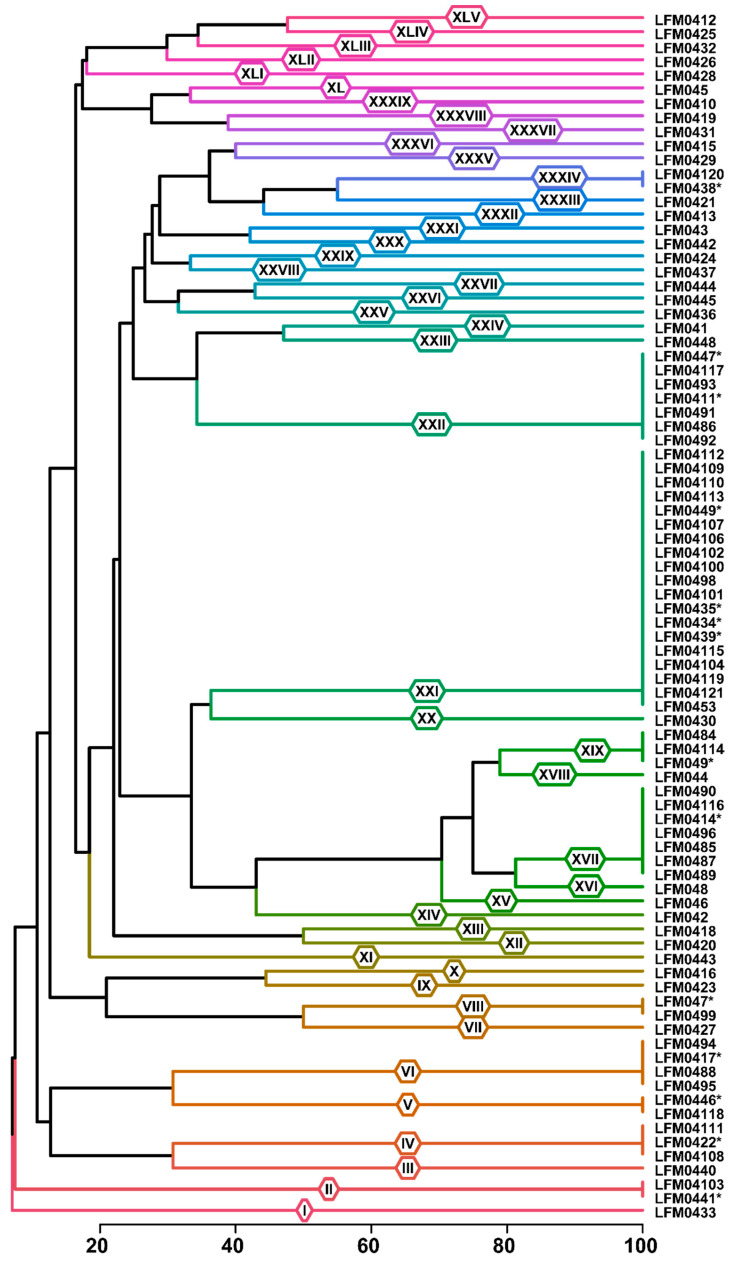
BOX-PCR dendrogram of *Fusarium* strains generated with Jaccard’s distance in the unweighted pair group method with arithmetic averages (UPGMA). The roman numerals on the branches represent the BOX-PCR groups. The scale represents the similarity in percentage and the BOX groups were defined on the basis of 100% similarity. Strains marked with asterisks were used in the pathogenicity tests in groups composed of more than one strain.

**Figure 5 jof-11-00230-f005:**
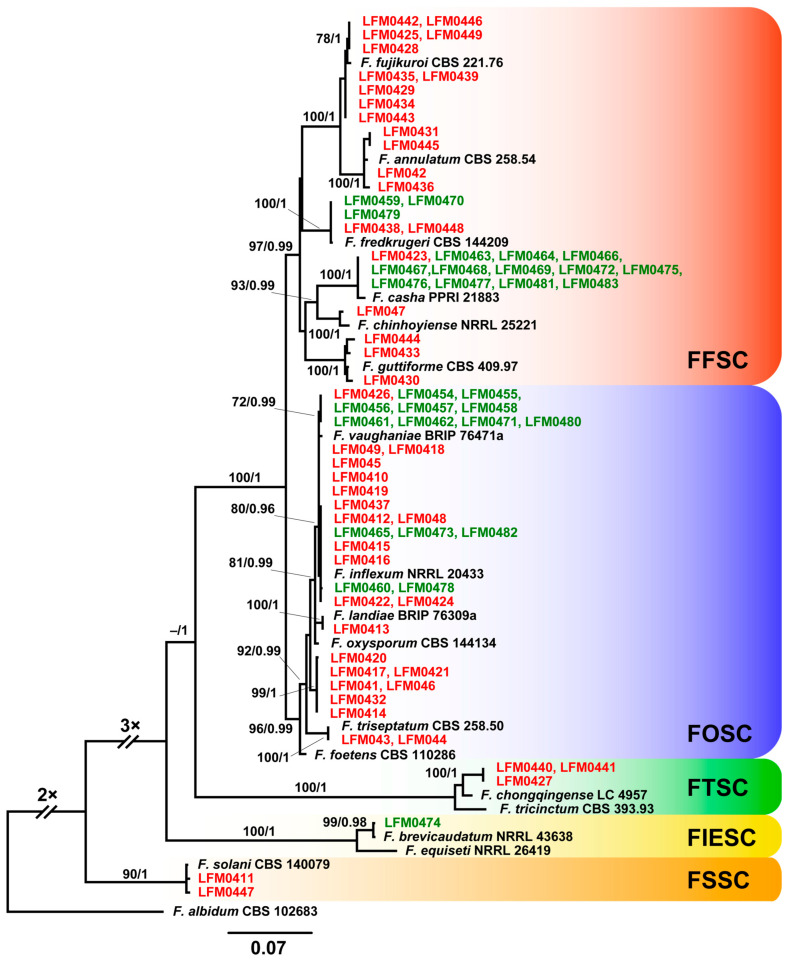
Phylogenetic tree of the *Fusarium* species inferred with the maximum likelihood (ML) and Bayesian inference (BI) methods with *tef1* sequences and the GTR + G model for both methods. The bootstrap analysis was performed with 1000 resamplings. The numbers on the nodes represent the bootstraps in ML analysis followed by the posterior probabilities (PP) in the Bayesian inference. Values lower than 70% in ML and lower than 0.95 PP in BI were not taken into consideration for branch support. Although the *tef1* gene of 79 strains was sequenced and all strains are shown in the phylogenetic tree, only one representative of identical sequences was deposited in the databases. Therefore, the evolutionary analysis involved a total of 65 distinct sequences with 673 bps of aligned nucleotides. Only sequences from type material available in the databases were used for comparison purposes and are listed in Appendix A. Strains from healthy plantlets are indicated in green and the ones from diseased plantlets are in red. *Fusarium albidum* (syn. *Luteonectria albida*) was used as an outgroup and the species complexes are indicated with different colours. The scale indicates the number of substitutions per site.

**Figure 6 jof-11-00230-f006:**
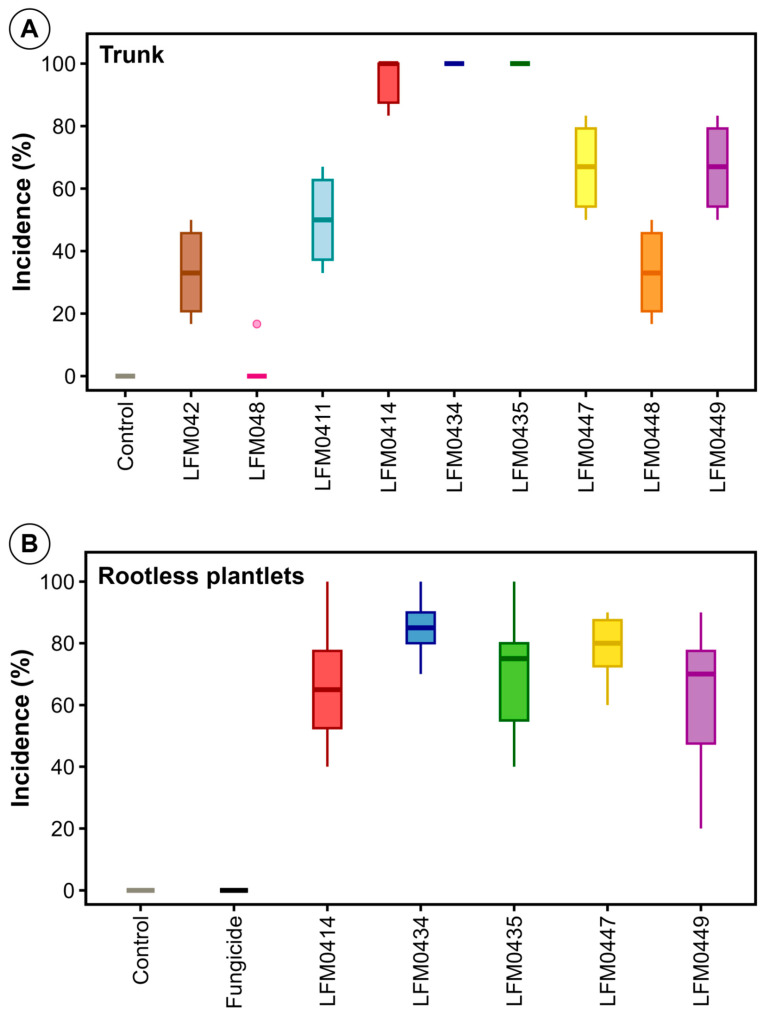
Trunk rot and root rot after cold storage in cv. Syrah grafted onto IAC766 rootstock. (**A**) Incidence of trunk rot. Plantlets were inoculated with a mycelial disc on the trunk wounded with a sterile cork borer and symptoms were evaluated 30 days later. (**B**) Incidence of root rot under cold storage. Rootless plantlets were inoculated with spore suspensions, grown in the field nursery for 12 months, removed from the soil, cleaned, and stored in a cold chamber for 3 months before evaluating the symptoms of root rot. The control treatments were non-inoculated plantlets and a fungicide treatment. Strain LFM0414 = *F. foetens*, LFM0434, LFM0435 and LFM0449 = *F. fujikuroi* and LFM0447 = *F. solani.* Data are averages of 100 plantlets per treatment.

**Figure 7 jof-11-00230-f007:**
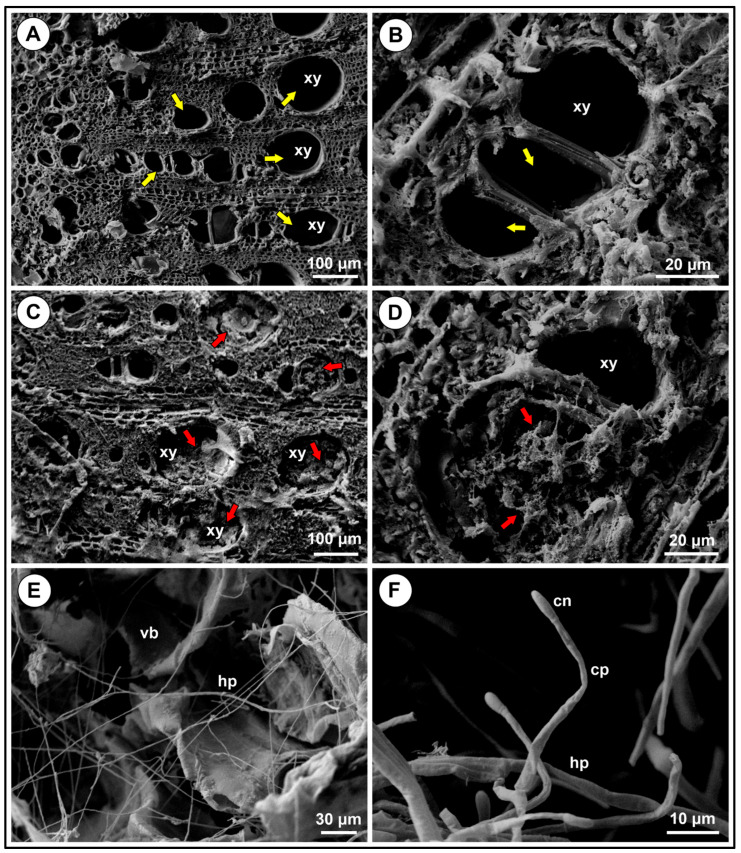
Scanning electron microscopy (SEM) images of xylem vessels of cv. Syrah grafted onto IAC766 rootstock. (**A**,**B**) Root xylem vessels of healthy plantlets showing the lumen of the vessels. (**C**,**D**) Root xylem of diseased plantlets. (**E**,**F**) Xylem vessels of diseased plantlets colonised by *Fusarium*. Abbreviations are xy = xylem, hp = hyphae, vb = vascular bundle, cn = conidium, and cp = conidiophore. Yellow arrows indicate the xylem in healthy tissues and red arrows show clogged xylem vessels in diseased plantlets. Scale bars are shown at the lower right corner of the figures.

**Figure 8 jof-11-00230-f008:**
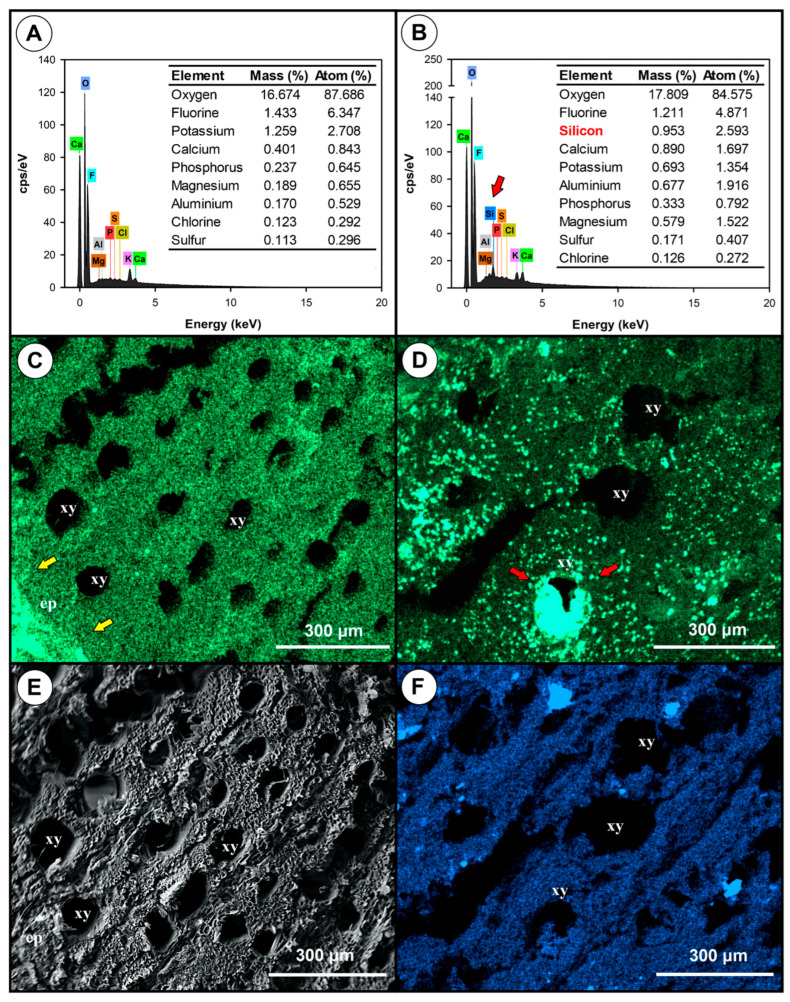
Energy-dispersive X-ray spectroscopy (EDS) of healthy and diseased root tissues of grapevine cv. Syrah grafted onto IAC766 rootstock. (**A**,**B**) EDS quantification of minerals in healthy and diseased tissues. (**C**,**D**) Distribution of Ca in healthy (**C**) and diseased (**D**) tissues; yellow arrows show concentration in the epidermis of healthy tissue and red arrows show concentration in the xylem. (**E**,**F**) Presence and distribution of Si in healthy and diseased tissue; lack of Si in healthy tissue (**E**) and Si evenly distributed in diseased tissue is indicated by the blue colour (**F**). Abbreviations are xy—xylem and ep—epidermis; scale bars are shown at the lower right corner of the microscopy figures.

**Table 1 jof-11-00230-t001:** Descriptive scale for evaluating the virulence of trunk rot in grapevine grafted plantlets infected with *Fusarium* in the pathogenicity assays with method (i). All strains listed in this study were tested with this method.

N° Diseased Plantlets	Pathogenicity/Virulence
0	-
<5	+
6–10	++
>11	+++

**Table 2 jof-11-00230-t002:** Identification and pathogenicity of *Fusarium* strains isolated from diseased roots of grapevine plantlets. The strains were selected based on their BOX groups, identification was performed by sequencing a fragment of the *tef1* gene, and pathogenicity was performed by the trunk inoculation method. Virulence was determined by quantifying the number of diseased inoculated plantlets according to a grading scale shown in Table 1. A total of 15 plantlets per treatment were evaluated 30 days after the inoculations. Identical accession numbers indicate that the strains had the same sequence, and in these cases only one representative sequence was deposited.

Strains	BOX-PCR Group	Closest Species (Type Material)	*tef1-α* (bp)	Accession Number	Identity (%)	Species Complex	Pathogenicity/Virulence ^1^
LFM0433	I	*F. guttiforme*	650	PV081890	99.23	FFSC	+++
LFM0441	II	*F*. *chongqingense*	645	PV081913	97.51	FTSC	+
LFM0440	III	*F*. *chongqingense*	645	PV081913	97.51	FTSC	+
LFM0422	IV	*F*. *inflexum*	651	PV081876	99.68	FOSC	++
LFM0446	V	*F. fujikuroi*	648	PV081893	99.53	FFSC	++
LFM0417	VI	*F*. *foetens*	658	PV081883	97.88	FOSC	+++
LFM0427	VII	*F*. *chongqingense*	639	PV081914	97.48	FTSC	+
LFM047	VIII	*F*. *chinhoyiense*	656	PV081900	98.92	FFSC	-
LFM0423	IX	*F*. *casha*	630	PV081901	99.21	FFSC	++
LFM0416	X	*F. inflfexum*	669	PV081877	100	FOSC	+++
LFM0443	XI	*F. fujikuroi*	639	PV081894	99.37	FFSC	+++
LFM0420	XII	*F*. *foetens*	668	PV081884	97.91	FOSC	++
LFM0418	XIII	*F. vaughaniae*	661	PV081870	99.54	FOSC	++
LFM042	XIV	*F. annulatum*	654	PV081904	99.54	FFSC	+
LFM046	XV	*F*. *foetens*	667	PV081885	98.20	FOSC	+++
LFM048	XVI	*F. inflexum*	671	PV081878	99.69	FOSC	+
LFM0414	XVII	*F*. *foetens*	673	PV081886	97.93	FOSC	+++
LFM044	XVIII	*F. triseptatum*	655	PV081888	99.51	FOSC	+++
LFM049	XIX	*F. vaughaniae*	661	PV081870	99.54	FOSC	++
LFM0430	XX	*F. guttiforme*	651	PV081891	99.23	FFSC	++
LFM0439	XXI	*F. fujikuroi*	656	PV081895	99.39	FFSC	++
LFM0434	XXI	*F. fujikuroi*	656	PV081896	99.39	FFSC	+++
LFM0435	XXI	*F. fujikuroi*	656	PV081895	99.39	FFSC	+++
LFM0449	XXI	*F. fujikuroi*	693	PV081897	99.54	FFSC	+++
LFM0411	XXII	*F. solani*	688	PV081912	99.41	FSSC	++
LFM0447	XXII	*F. solani*	706	PV081911	99.86	FSSC	++
LFM0448	XXIII	*F. fredkrugeri*	706	PV081908	98.67	FFSC	+++
LFM041	XXIV	*F*. *foetens*	667	PV081885	98.20	FOSC	++
LFM0436	XXV	*F. annulatum*	654	PV081905	98.93	FFSC	++
LFM0445	XXVI	*F. annulatum*	658	PV081906	99.09	FFSC	+
LFM0444	XXVII	*F. guttiforme*	653	PV081892	98.47	FFSC	+++
LFM0437	XXVIII	*F. inflexum*	651	PV081879	99.68	FOSC	++
LFM0424	XXIX	*F*. *inflexum*	651	PV081876	99.68	FOSC	++
LFM0442	XXX	*F. fujikuroi*	648	PV081893	99.53	FFSC	+++
LFM043	XXXI	*F. triseptatum*	655	PV081888	99.51	FOSC	++
LFM0413	XXXII	*F. landiae*	664	PV081889	100	FOSC	++
LFM0421	XXXIII	*F*. *foetens*	658	PV081883	97.88	FOSC	++
LFM0438	XXXIV	*F. fredkrugeri*	706	PV081908	98.67	FFSC	+
LFM0429	XXXV	*F. fujikuroi*	644	PV081898	99.38	FFSC	++
LFM0415	XXXVI	*F. inflexum*	653	PV081880	100	FOSC	+
LFM0431	XXXVII	*F*. *annulatum*	645	PV081907	99.07	FFSC	+
LFM0419	XXXVIII	*F. vaughaniae*	661	PV081871	99.69	FOSC	++
LFM0410	XXXIX	*F. vaughaniae*	639	PV081872	99.69	FOSC	+
LFM045	XL	*F. vaughaniae*	650	PV081873	99.69	FOSC	+++
LFM0428	XLI	*F. fujikuroi*	639	PV081899	99.53	FFSC	+
LFM0426	XLII	*F. vaughaniae*	660	PV081869	99.54	FOSC	+++
LFM0432	XLIII	*F*. *foetens*	655	PV081887	97.87	FOSC	+
LFM0425	XLIV	*F. fujikuroi*	693	PV081897	99.54	FFSC	+
LFM0412	XLV	*F. inflexum*	671	PV081878	99.69	FOSC	-

^1^ Defined according to Table 1. FOSC = *Fusarium oxysporum* species complex, FTSC = *Fusarium tricinctum* species complex, and FSSC = *Fusarium solani* species complex.

**Table 3 jof-11-00230-t003:** Colonisation and pathogenicity of *Fusarium* strains obtained from diseased grapevine plantlets. These strains were inoculated on surface-sterilised roots of grapevine plantlets and seedlings of common bean and tomato, and after a period of 7 days of incubation these roots were surface-sterilised again and plated on culture media to determine the percentage of re-isolation. Disease symptoms were analysed visually on the roots. Controls were non-inoculated plantlet roots. The absence of symptoms is indicated by the sign “–”.

Strains ^1^	BOX-PCR Group	Closest Species (Type Material)	Grapevine		Tomato		Common Bean	
			Re-Isolation (%)	Symptoms	Re-Isolation (%)	Symptoms	Re-Isolation (%)	Symptoms
LFM042	XIV	*F. annulatum*	100.0	–	94.0	–	88.0	–
LFM048	XVI	*F. inflexum*	91.6	–	92.0	–	89.0	–
LFM0411	XXII	*F. solani*	75.0	–	95.0	–	97.0	–
LFM0414	XVII	*F. foetens*	91.6	–	93.0	–	94.0	–
LFM0434	XXI	*F. fujikuroi*	100.0	–	96.0	–	93.0	–
LFM0435	XXI	*F. fujikuroi*	100.0	–	100.0	–	91.0	–
LFM0447	XXII	*F. solani*	75.0	–	92.0	–	92.0	–
LFM0448	XXIII	*F. fredkrugeri*	91.6	–	96.0	–	87.0	–
LFM0449	XXI	*F. fujikuroi*	91.6	–	90.0	–	85.0	–
Control	-	-	0.0	–	0.0	–	0.0	–

**Table 4 jof-11-00230-t004:** *Fusarium* strains isolated from different parts of healthy plantlets. The *tef1* gene was sequenced for all strains, and identical sequences as indicated by the same length in base pairs (bps) and by the same sequence identity (%) were deposited only once, and, therefore, received only one accession number. Pathogenicity and virulence were determined by the trunk inoculation method with 15 plantlets per treatment and evaluations performed 30 days after the inoculations by using the descriptive scale shown in Table 1.

Strains	Isolation Source	Species (Type Material)	*tef1-α* (bp)	Accession Number	Identity (%)	Species Complex	Pathogenicity/Virulence ^1^
LFM0464	Trunk	*F. casha*	657	PV081902	99.39	FFSC	++
LFM0466	Trunk	*F. casha*	657		99.39	FFSC	+
LFM0468	Root	*F. casha*	657		99.39	FFSC	++
LFM0469	Trunk	*F. casha*	657		99.39	FFSC	+
LFM0472	Root	*F. casha*	657		99.39	FFSC	+++
LFM0475	Trunk	*F. casha*	657		99.39	FFSC	-
LFM0476	Root	*F. casha*	657		99.39	FFSC	++
LFM0477	Root	*F. casha*	657		99.39	FFSC	++
LFM0481	Trunk	*F. casha*	657		99.39	FFSC	+
LFM0483	Trunk	*F. casha*	657		99.39	FFSC	+
LFM0463	Root	*F. casha*	654	PV081903	99.38	FFSC	+++
LFM0467	Trunk	*F. casha*	654		99.38	FFSC	-
LFM0459	Trunk	*F. fredkrugeri*	658	PV081909	98.62	FFSC	+++
LFM0470	Trunk	*F. fredkrugeri*	658		98.62	FFSC	++
LFM0479	Trunk	*F. fredkrugeri*	655	PV081910	98.61	FFSC	++
LFM0460	Root	*F. inflexum*	581	PV081881	99.66	FOSC	++
LFM0478	Root	*F. inflexum*	581		99.66	FOSC	++
LFM0462	Root	*F*. *vaughaniae*	660	PV081874	99.54	FOSC	-
LFM0454	Root	*F*. *vaughaniae*	657	PV081875	99.54	FOSC	-
LFM0455	Root	*F*. *vaughaniae*	657		99.54	FOSC	++
LFM0456	Root	*F*. *vaughaniae*	657		99.54	FOSC	-
LFM0457	Root	*F*. *vaughaniae*	657		99.54	FOSC	+
LFM0458	Root	*F*. *vaughaniae*	657		99.54	FOSC	-
LFM0461	Root	*F*. *vaughaniae*	657		99.54	FOSC	++
LFM0471	Root	*F*. *vaughaniae*	657		99.54	FOSC	++
LFM0480	Root	*F*. *vaughaniae*	657		99.54	FOSC	++
LFM0465	Root	*F. inflexum*	613	PV081882	100	FOSC	+
LFM0473	Root	*F. inflexum*	613		100	FOSC	+
LFM0482	Root	*F*. *inflexum*	613		100	FOSC	+++
LFM0474	Trunk	*F*. *brevicaudatum*	651	PV081915	99.68	FIESC	-

^1^ Defined according to Table 1. FFSC = *Fusarium fujikuroi* species complex, FOSC = *Fusarium oxysporum* species complex, and FIESC = *Fusarium incarnatum-equiseti* species complex.

## Data Availability

The original contributions presented in the study are included in the article, further inquiries can be directed to the corresponding author.

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
