# Peer review of "A New Root and Trunk Rot Disease of Grapevine Plantlets Caused by Fusarium in Four Species Complexes"

_jof, 2025, doi:10.3390/jof11030230_

Round 1

Reviewer 1 Report

 This manuscript provides a well-structured and in-depth analysis of a novel root and trunk rot disease affecting grapevine planting material under cold storage, caused by Fusarium species from four species complexes. The study employs robust methodology, including pathogenicity assays, genetic diversity studies (BOX-PCR), molecular identification (tef1 gene sequencing), and microscopic and spectroscopic analyses to elucidate pathogen-host interactions.

 The research is highly relevant to plant pathology and viticulture, offering new insights into disease development and resistance responses. However, certain areas require clarification and improvement, particularly in methodological details, data interpretation, and discussion structure.

The title and several section headings are a bit too long and could be reworded for conciseness. The introduction provides a strong background but should better emphasize the study’s novelty and how this disease differs from previously described GTD in grapevine.

The methodology section requires clearer details on the number of symptomatic plants collected, the isolation process, and the justification for using BOX-PCR over multilocus sequence typing (MLST). Additional loci (rpb2, calmodulin) could strengthen species delineation.

The pathogenicity assays, though well-described, could be made clearer and more concise, with statistical analysis details explicitly stated.

The discussion is informative but contains some redundancy, which should be addressed for a more structured and impactful presentation of findings.

Figure legends should clearly explain symbols and scales, and terminology should be standardized, avoiding the alternating use of “plantlets” and “grapevine planting material.” Additionally, the manuscript contains complex and lengthy sentences that could be simplified for improved readability. Overall, this study presents valuable findings and is methodologically sound, but revisions should focus on conciseness, clarification of methods, and reducing repetition.

Mentioned above

Author Response

Dear Reviewer 1,

 We thank you for the thorough review that certainly improved the quality of our work. Below you will find our point by point answers to your comments and suggestions.

We prepared two files for this review, one called JoF_Mn_marked_version containing all changes we have made following the reviewers suggestions and the other version, named JoF_Mn_unmarked_version, where we removed all the markings made in the marked version for typesetting purposes. In the marked version, all comments and suggestions made by reviewer 1 are highlighted in yellow.

Our answers to your questions/suggestions are presented immediately following each comment.

Comment 1: This manuscript provides a well-structured and in-depth analysis of a novel root and trunk rot disease affecting grapevine planting material under cold storage, caused by Fusarium species from four species complexes. The study employs robust methodology, including pathogenicity assays, genetic diversity studies (BOX-PCR), molecular identification (tef1 gene sequencing), and microscopic and spectroscopic analyses to elucidate pathogen-host interactions.

The research is highly relevant to plant pathology and viticulture, offering new insights into disease development and resistance responses. However, certain areas require clarification and improvement, particularly in methodological details, data interpretation, and discussion structure.

Response 1: We thank the reviewer for the positive feedback on our manuscript.

Comment 2: The title and several section headings are a bit too long and could be reworded for conciseness.

Response 2: We agree with reviewer 1 that our title is a little long, but it expresses the main findings of our work and catches the attention of readers. Additionally, it is not uncommon to find even longer titles, which may be interesting as frequently conciseness comes with a cost on clarity. Our subheadings are not longer than a line and therefore should not be considered too long and as we argued above for the title, they are informative. We respectfully argue that it is not necessary to compress our title and subheadings as it would impair our manuscript’s clarity.

Comment 3: The introduction provides a strong background but should better emphasize the study’s novelty and how this disease differs from previously described GTD in grapevine.

Response 3: We agree with the reviewer and added more information that clearly states that root and trunk rot differs from GTD. Please, see lines 79-80 of the marked version. 

Comment 4: The methodology section requires clearer details on the number of symptomatic plants collected, the isolation process, and the justification for using BOX-PCR over multilocus sequence typing (MLST). Additional loci (rpb2, calmodulin) could strengthen species delineation.

Response 4: We thank the reviewer for these comments. We indeed did not have the information on the number of plantlets used in the isolations in the material and methods section, but we had it in the legend of Figure 3. We added the information requested in material and methods, please, see lines 89-91. The isolation process was already described, please, see lines 91-97.

There is no relation between MLST and BOX-PCR. MLST involves sequencing and analysis of specific regions of the genome and is useful in phylogenetic analysis and molecular identification, whereas BOX-PCR is a whole-genome fingerprinting technique used in diversity studies with no relation to phylogeny and identification. We added a sentence on the reasons for the use of BOX-PCR, please, see lines 106-107.

Sequences of tef1 are sufficient to identify Fusarium at the species level, which is our objective in this study. If our objective were to describe novel species, we would certainly use other regions, including the ones suggested by the reviewer. The following publication shows that tef1 is the most variable and therefore the most useful among the markers used for Fusarium identification: https://doi.org/10.1094/PDIS-09-21-2035-SR

Comment 5: The pathogenicity assays, though well-described, could be made clearer and more concise, with statistical analysis details explicitly stated.

Response 5: We thank the reviewer for this comment and we agree that this section is very detailed. As it involves the description of three methods, details are essential to guarantee its reproduction by the interested readership. Lack of details would compromise its clarity. The statistical analyses are clearly described in lines 172-175.

Comment 6: The discussion is informative but contains some redundancy, which should be addressed for a more structured and impactful presentation of findings.

Response 6: We agree with the reviewer and strove to remove the redundant text. We deleted lines 472-477 and 586-589 that contained repeated information.

Comment 7: Figure legends should clearly explain symbols and scales, and terminology should be standardized, avoiding the alternating use of “plantlets” and “grapevine planting material.”

Response 7: We thank the reviewer for the comment. We reviewed all figures and modified Figures 2, 7 and 8, where we indicated where the scale bars are located. Please, see lines 252-253, 396 and 404-405. We also choose the word plantlet over planting material to avoid confusion.

Comment 8: Additionally, the manuscript contains complex and lengthy sentences that could be simplified for improved readability. Overall, this study presents valuable findings and is methodologically sound, but revisions should focus on conciseness, clarification of methods, and reducing repetition.

Response 8: We thank the reviewer very much for these comments. We already dealt with the methodological details and reduced repetition in our previous responses, please see above. We modified a long sentence in the discussion section, splitting it into 3 sentences to facilitate readability and understanding, please see lines 562-567 of the marked version.

Reviewer 2 Report

the manuscript "The New Root and Trunk Rot Disease of Grapevine Planting Material that Manifests Itself Under Cold Storage is Caused by Fusarium in Four Species Complexes" describes what  they called grapevine trunk decline (GTD) and identify the agents of this disease focusing on Fusarium species. They collected Fusarium spp. from both healthy and disesed plants and identified the species using tef1 sequencing. They further analysed the symptoms using SEM and concluded that Fusarium spp. may be acting as opportunistic pathogens especially in the cold storage stage of the grapevine plantations. The manuscript is logically written and relevant to the field of fungal pathogens. I only have minor comments.

231-232: Figure 1D is mentioned later than Figure 1 E and F. Please place D after E and F in the Figure 1.

249: "Asterisks indicate significant differences according to the t-test at 5% probability." What does ** (2 asterisks) mean?

251-253: This can be a figure.

260: Did you mean "(Figure 2B, C, E, F)"?

357:  please add "by all strains at various degrees" since LFM048 shows very little disease symptom.

361: Please explain why you did not use all the strains in Figure 6A in Figure 6B.

For all figures please write "healthy" and "diseased" on the figure so that the readers do not solely depend on the figure legends to understand.

Author Response

Dear Reviewer 2,

We thank you for the thorough review that certainly improved the quality of our work. Below you will find our point by point answers to your comments and suggestions.

We prepared two files for this review, one called JoF_Mn_marked_version containing all changes we have made following the reviewers suggestions and the other version, named JoF_Mn_unmarked_version, where we removed all the markings made in the marked version for typesetting purposes. In the marked version, all comments and suggestions made by reviewer 2 are highlighted in green.

Our answers to your questions/suggestions are presented immediately following each comment.

Comment 1: The manuscript "The New Root and Trunk Rot Disease of Grapevine Planting Material that Manifests Itself Under Cold Storage is Caused by Fusarium in Four Species Complexes" describes what  they called grapevine trunk decline (GTD) and identify the agents of this disease focusing on Fusarium species. They collected Fusarium spp. from both healthy and diseased plants and identified the species using tef1 sequencing. They further analysed the symptoms using SEM and concluded that Fusarium spp. may be acting as opportunistic pathogens especially in the cold storage stage of the grapevine plantations. The manuscript is logically written and relevant to the field of fungal pathogens. I only have minor comments.

Response 1: We thank the reviewer for the positive comments on our manuscript.

Comment 2: 231-232: Figure 1D is mentioned later than Figure 1 E and F. Please place D after E and F in the Figure 1.

Response 2: We thank the reviewer for the thorough review and for noticing this problem, which we corrected in the reviewed version. Please, see line 234 of the marked version.  

Comment 3: 249: "Asterisks indicate significant differences according to the t-test at 5% probability." What does ** (2 asterisks) mean?

Response 3: We thank the reviewer for this observation. The t-tests were done with 5 and 1% probability and they were significant at 1%, which is indicated by two asterisks. We correctly indicated the level of significance in the figures and also indicated in the text that the analyses were done at 1 and 5%. Please, see lines 175 and 252.

Comment 4: 251-253: This can be a figure.

Response 4: We thank the reviewer for this suggestion. However, a figure with only two bars, one showing 100% of plant death and the other bar showing 0% of plant death, both without any variance is not very informative and is easily described in the text. Therefore, we would prefer to leave it the way it is. Besides that, the description in the text also contributes to decrease the number of figures, making the text more concise as requested by reviewer 1.  

Comment 5: 260: Did you mean "(Figure 2B, C, E, F)"?

Response 5: We thank the reviewer for the comment. We corrected it as suggested, please, see lines 258-259 and 263 of the marked version.

Comment 6: 357:  please add "by all strains at various degrees" since LFM048 shows very little disease symptom.

Response 6: We thank the reviewer for this suggestion. We added the information on line 357.

Comment 7: 361: Please explain why you did not use all the strains in Figure 6A in Figure 6B.

Response 7: We thank the reviewer for this comment as it gave us the opportunity to improve the text. We had an explanation for that in the discussion, see lines 491-497. However, as the reviewer pointed out it was missing in the results section. We added an explanation for this selection in the results section, please, see lines 361-362 of the marked version.

Comment 8: For all figures please write "healthy" and "diseased" on the figure so that the readers do not solely depend on the figure legends to understand.

Response 8: We agree with the reviewer, but unfortunately there is no space for these words in figure 2D. Therefore, we described the abbreviations of healthy and diseased in Figure 2A and maintained H and D in Figure 2D. These abbreviations are intuitive and easily understood by readers.